# Distilling Representations from GAN Generator via Squeeze and Span

**Yu Yang**[1*], **Xiaotian Cheng**[1*], **Chang Liu**[1], **Hakan Bilen**[2], **Xiangyang Ji**[1]
[1]Tsinghua University, BNRist    [2]University of Ediburgh
yang-yu16@foxmail.com, cxt20@mails.tsinghua.edu.cn
{liuchang2022, xyji}@tsinghua.edu.cn, hbilen@ed.ac.uk

## Abstract

In recent years, generative adversarial networks (GANs) have been an actively studied topic and shown to successfully produce high-quality realistic images in various domains. The controllable synthesis ability of GAN generators suggests that they maintain informative, disentangled, and explainable image representations, but leveraging and transferring their representations to downstream tasks is largely unexplored. In this paper, we propose to distill knowledge from GAN generators by squeezing and spanning their representations. We *squeeze* the generator features into representations that are invariant to semantic-preserving transformations through a network before they are distilled into the student network. We *span* the distilled representation of the synthetic domain to the real domain by also using real training data to remedy the mode collapse of GANs and boost the student network performance in a real domain. Experiments justify the efficacy of our method and reveal its great significance in self-supervised representation learning. Code is available at https://github.com/yangyu12/squeeze-and-span.

## 1   Introduction

Generative adversarial networks (GANs) [23] continue to achieve impressive image synthesis results thanks to large datasets and recent advances in network architecture design [5, 36, 37, 34]. GANs synthesize not only realistic images but also steerable ones towards specific content or styles [22, 52, 49, 33, 57, 53, 32]. These properties motivate a rich body of works to adopt powerful pretrained GANs for various computer vision tasks, including part segmentation [68, 56, 61], 3D reconstruction [67], image alignment [48, 45], showing the strengths of GANs in the few-label regime.

GANs typically produce fine-grained, disentangled, and explainable representations, which allow for higher data efficiency and better generalization [42, 68, 56, 61, 67, 48]. Prior works on GAN-based representation learning focus on learned features from either a discriminator network [50] or an encoder network mapping images back into the latent space [19, 17, 18]. However, there is still inadequate exploration about how to leverage or transfer the learned representations in *generators*. Inspired by the recent success of [68, 56, 61], we hypothesize that representations produced in generator networks are rich and informative for downstream discriminative tasks. Hence, this paper proposes to distill representations from feature maps of a pretrained generator network into a student network (see Fig. 1).

In particular, we present a novel "squeeze-and-span" technique to distill knowledge from a generator into a representation network[2] that is transferred to a downstream task. Unlike transferring discrimi-

---

[*]Equal Contribution

[2]Throughout the paper, two terms"representation network" and "student network" are used interchangeably, as are the "generator network" and "teacher network".

36th Conference on Neural Information Processing Systems (NeurIPS 2022).

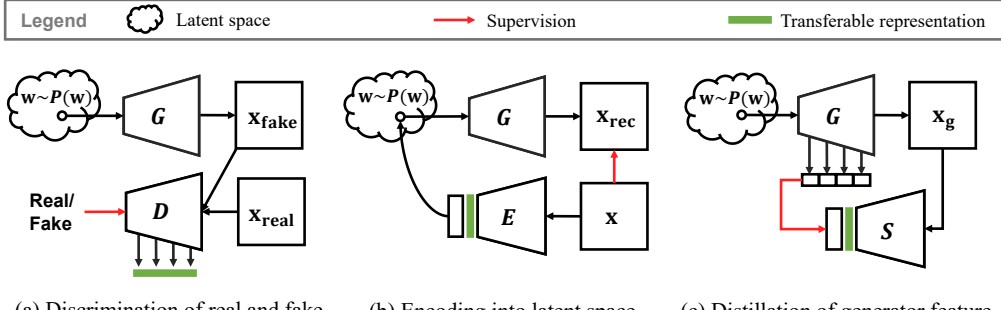

(a) Discrimination of real and fake    (b) Encoding into latent space    (c) Distillation of generator feature

Figure 1: A comparison of representation transfer in GANs. (a) Transferring representations in discriminator ($D$) which is tasked to distinguish real or fake images (*e.g.* [50]). (b) Transferring representations in encoder ($E$) which projects an image into latent space (*e.g.* [19, 17, 18]). (c) Transferring representations in student ($S$) which predicts the generator features (ours).

nator network, generator network is *not* directly transferable to downstream image recognition tasks, as it cannot ingest image input but a latent vector. Hence, we distill generator network representations into a representation network that can be further transferred to the target task. When fed in a synthesized image, the representation network is optimized to produce similar representations to the generator network's. However, the generator representations are very high-dimensional and not all of them are informative for the downstream task. Thus, we propose a squeeze module that purifies generator representations to be invariant to semantic-preserving transformations through an MLP and an augmentation strategy. As the joint optimization of the squeeze module and representation network can lead to a trivial solution (*e.g.* mapping representations to zero vector), we employ variance-covariance regularization in [3] while maximizing the agreement between the two networks. Finally, to address the potential domain gap between synthetic and real images, we span the learned representation of synthetic images by training the representation network additionally on real images.

We evaluate our distilled representations on CIFAR10, CIFAR100 and STL10 with linear classification tasks as commonly done in representation learning. Experimental results show that squeezing and spanning generator representations outperforms methods that build on discriminator and encoding images into latent space. Moreover, our method achieves better results than discriminative SSL algorithms, including SimSiam [10] and VICReg [3] on CIFAR10 and CIFAR100, and competitive results on STL10, showing significant potential for transferable representation learning.

Our contributions can be summarized as follows: We (1) provide a new taxonomy of representation and transfer learning in generative adversarial networks based on the location of the representations, (2) propose a novel "squeeze-and-span" framework to distill representations in the GAN generator and transfer them for downstream tasks, (3) empirically show the promise of utilizing generator features to benefit self-supervised representation learning.

## 2    Related Work

**GANs for Representation Learning.**    Significant progress has been made on the interpretability, manipulability, and versatility of the latent space and representation of GANs [36, 37, 34, 35]. It inspires a broad spectrum of GAN-based applications, such as semantic segmentation [68, 56, 61], visual alignment [48, 45], and 3D reconstruction [67], where GAN representations are leveraged to synthesize supervision signals efficiently. As GAN can be trained unsupervised, its representations are transferred to downstream tasks. DCGAN [50] proposes a convolutional GAN and uses the pre-trained discriminator for image classification. BiGAN [17] adopts an inverse mapping strategy to transfer the real domain knowledge for representation learning. While ALI [19] improves this idea with a stochastic network instead of a deterministic one, BigBiGAN [18] extends BiGAN with BigGAN [5] for large scale representation learning. GHFeat [59] trains a post hoc encoder that maps given images back into style codes of style-based GANs [36, 37, 35] for image representation. These works leverage or transfer representations from either discriminators or encoders. In contrast, our method reveals that the generator of a pre-trained GAN is typically more suitable for representation transfer with a proper distillation strategy.

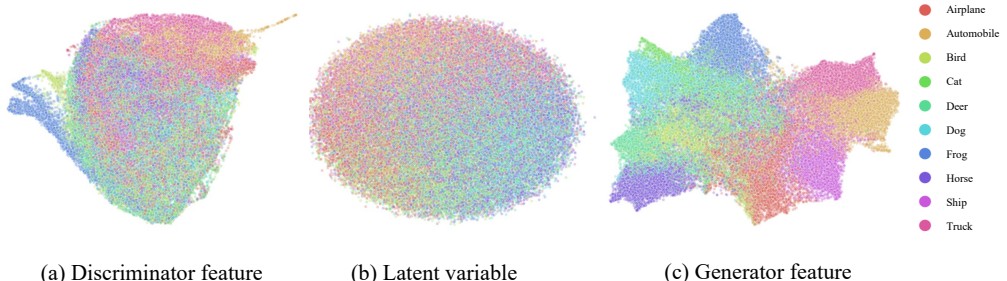

| | |
|---|---|
| (a) Discriminator feature | (b) Latent variable | (c) Generator feature |

Figure 2: Visualization of three types of GAN representations: (a) discriminator feature, (b) latent variable, and (c) generator feature. An unconditional StyleGAN2-ADA model pre-trained on CIFAR10 is employed. Colors indicate different classes. Generator features are naturally clustered and consistent with classes.

**Knowledge Distillation (KD)**   aims at training a small student network, under the supervision of a relatively large teacher network [31]. In terms of the knowledge source, it can be broadly divided into logit-based KD and feature-based KD. Logit-based KD methods [41, 60, 12] optimize the divergence loss between the predicted class distributions, usually called logits or soft labels, of the teacher and student network. Feature-based KD methods [38, 2, 54] adopt the teacher model's intermediate layers as supervisory signals for the student. FitNet [51] introduces the output of hidden layers of the teacher network as supervision. AT [63] proposes to match attention maps between the teacher and student. FSP [62] calculates flow between layers as guidance for distillation. Likewise, our method distills knowledge from intermediate layers from a pre-trained GAN generator.

**Self-Supervised Representation Learning (SSL)**   pursues learning general transferable representations from unlabelled data. To produce informative self-supervision signals, the design of handcrafted pretext tasks has flourished for a long time, including jigsaw puzzle completion [46], relative position prediction [15, 16], rotation perception [21], inpainting [47], colorization [40, 65], masked image modeling [27, 58], *etc*. Instead of performing intra-instance prediction, contrastive learning-based SSL methods explore inter-instance relation. Applying the InfoNCE loss or its variants [26], they typically partition informative positive/negative data subsets and attempt to attract positive pairs while repelling negative ones. MoCo series [28, 8, 11] introduce an offline memory bank to store large negative samples for contrast and a momentum encoder to make them consistent. SimCLR [7] adopts an end-to-end manner to provide negatives in a mini-batch and introduce substantial data augmentation and a projection head to improve the performance significantly. Surprisingly, without negative pairs, BYOL [25] proposes a simple asymmetry SSL framework with the momentum branch applying the stop gradient to avoid model collapse. It inspires a series of in-deep explorations, such as SimSiam [9], Barlow Twins [64], VICReg [3], *etc*. In this paper, despite the same end goal of obtaining transferable representations and the use of techniques from VICReg [3], we study the transferability of generator representations in pretrained GANs to discriminative tasks, use asymmetric instead of siamese networks, and design effective distillation strategies.

## 3   Rethinking GAN Representations

Let $G : \mathcal{W} \to \mathcal{X}$ denote a generator network that maps a latent variable in $\mathcal{W}$ to an image in $\mathcal{X}$. An *unconditional* GAN trains $G$ adversarially against a discriminator network $D : \mathcal{X} \to [0, 1]$ that estimates the realness of the given images,

$$\max_{G} \min_{D} \mathbb{E} \log(1 - D(G(\mathbf{w}))) + \log D(\mathbf{x}). \tag{1}$$

The adversarial learning does not require any human supervision and therefore allows for learning representations in an unsupervised way. In this paper, we show that the type of GAN representations and how they are obtained has a large effect on their transferrability. To illustrate the impact on the transferability, Fig. 2 plots the embedded 2D points of three different type of representations from an

unconditional GAN, where color is assigned based on the class labels.[3] Note that we describe each representation in the following paragraphs.

**Discriminator Feature**    The discriminator $D$, which is tasked to distinguish real and fake images, can be transferred to various recognition tasks [50]. Formally, let $D = d^{(L)} \circ d^{(L-1)} \circ \cdots \circ d^{(1)}$ denote the decomposition of a discriminator into $L$ consecutive layers. As shown in Fig. 1(a), given an image $\mathbf{x}$, the discriminator representation can be extracted by concatenating the features after average pooling from each discriminator block output,

$$\mathbf{h}^d = [\mu(\mathbf{h}^d_1), \ldots, \mu(\mathbf{h}^d_L)], \quad \text{where } \mathbf{h}^d_i = d^{(i)} \circ \cdots \circ d^{(1)}(\mathbf{x}), \tag{2}$$

where $\mu$ denotes the average pooling operator. However, Fig. 2(a) shows that the cluster of discriminator features is not significantly correlated with class information indicating that real/fake discrimination does not necessarily relate to class separation.

**Latent Variable**    An alternative way of transferring GAN representation is through its latent variable $\mathbf{w}$ [19, 17, 18]. In particular, one can invert the generator such that it can extract a latent variable representation of the generated image through a learned encoder $E$. Then the representations of the encoder can be transferred to a downstream task. While some works jointly trains the encoder with the generator and discriminator [19, 17, 18], we consider training a *post hoc* encoder [6] given a fixed pre-trained generator $G$, as this provides more consistent comparison with the other two strategies:

$$E^* = \arg\min_E \mathbb{E}_{\mathbf{w} \sim P(\mathbf{w}), \mathbf{x} = G(\mathbf{w})} \left[ \|G(E(\mathbf{x})) - \mathbf{x}\|_1 + \mathcal{L}_{\text{percep}}(G(E(\mathbf{x})), \mathbf{x}) + \lambda \|E(\mathbf{x}) - \mathbf{w}\|_2^2 \right],$$
$$\tag{3}$$

where $\mathcal{L}_{\text{percep}}$ denotes the LPIPS loss [66] and $\lambda = 1.0$ is used to balance different loss terms. The key assumption behind this strategy is that latent variables encode various characteristics of generated images (*e.g.* [33, 57]) and hence extracting them from generated images result in learning transferrable representations.

Fig. 2(b) visualizes the embedding of latent variables[4]. It shows that samples from the same classes are not clustered together and distant from other ones. In other words, latent variables do not disentangle the class information while encoding other information about image synthesis.

**Generator Feature**    An overlooked practice is to utilize generator features. Typically, GAN generators transform a low-resolution (*e.g.* 4×4) feature map to a higher-resolution one (*e.g.* 256×256) and further synthesize images from the final feature map [17, 36] or multi-scale feature maps [37]. The image synthesis is performed hierarchically: feature map from low to high resolution encodes the low-frequency to high-frequency component for composing an image signal [35]. This understanding is also evidenced by image editing works [22, 52, 49, 53, 32] which show that interfering with low-resolution feature maps leads to a structural and high-level change of an image, and altering high-resolution feature maps only induces subtle appearance changes. Therefore, generator features contain valuable hierarchical knowledge about an image. Formally, let $G = g^{(L)} \circ g^{(L-1)} \circ \cdots \circ g^{(1)}$ denote the decomposition of a discriminator into $L$ consecutive layers. Given a latent variable $\mathbf{w} \sim P(\mathbf{w})$ drawn from a prior distribution, we consider the concatenated features average pooled from each generator block output,

$$\mathbf{h}^g = [\mu(\mathbf{h}^g_1), \ldots, \mu(\mathbf{h}^g_L)], \quad \text{where } \mathbf{h}^g_i = g^{(i)} \circ \cdots \circ g^{(1)}(\mathbf{w}). \tag{4}$$

As Fig. 2(c) shows, generator features within the same class are naturally clustered. This result suggests that generators contain identifiable representations that can be transferred for downstream tasks. However, as GANs do not initially provide a reverse model for the accurate recovery of generator features, it is still inconvenient to extract generator features for any given image. This limitation motivates us to distill the valuable features from GAN generators.

---

[3]The GAN is trained trained on CIFAR10. We use UMAP embeddings [44] for dimensionality reduction. As the class labels of the generated images are unknown, they are inferred by a classifier that is trained on CIFAR10 training set and achieves around 95% top-1 accuracy on CIFAR10 validation set.

[4]In the family of StyleGAN, to achieve more disentangled latent variables, the prior latent variable that observes standard normal distribution is mapped into a learnable latent space via an MLP before fed into the generator. In our work, we refer to latent variables as the transformed ones, which are also known as latent variables in $\mathcal{W}^+$ space in other works [1]

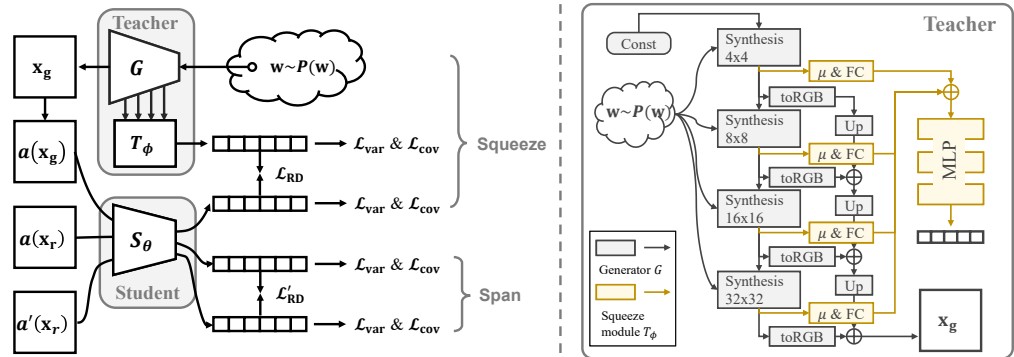

Figure 3: Squeeze and span representation from the GAN generator. *Left*: pretrained generator $G$ and squeeze module $T_\phi$ constitute teacher network to produce squeezed representations which are further distilled into a student network $S_\theta$ (Squeeze part). The student network is also trained on real data (Span part). *Right*: the generator structure and our squeeze module. We average pool (denoted as $\mu$) the feature maps from each synthesis block and transform them with a linear layer plus an MLP, termed squeeze module.

## 4 Squeeze-and-Span Representations from GAN Generator

This section introduces the "Squeeze-and-Span" technique to distill representation from GANs into a student network, which can then be readily transferred for downstream tasks, *e.g.* image classification. Let $S_\theta : \mathcal{X} \to \mathcal{H}$ denote a student network that maps a given image into representation space. A naive way of representation learning can be achieved by tasking the student network to predict the teacher representation, which can be formulated as the following optimization problem:

$$\min_\theta \; \mathbb{E}_{\mathbf{w} \sim P(\mathbf{w})} \, \|S_\theta(G(\mathbf{w})) - \mathbf{h}^g(\mathbf{w})\|_2^2, \qquad (5)$$

where we use mean squared error to measure the prediction loss and $\mathbf{h}^g(\mathbf{w})$ to denote the dependence of $\mathbf{h}^g$ on $\mathbf{w}$. However, this formulation has two problems. First, representations extracted through multiple layers of the generator are likely to contain significantly redundant information for downstream tasks but necessary for image synthesis. Second, as the student network is only optimized on synthetic images, it is likely to perform poorly in extracting features from real images in the downstream task due to the potential domain gap between real and synthetic images. To mitigate these issues, we propose the "Squeeze and Span" technique as illustrated in Fig. 3.

### 4.1 Squeezing Informative Representations

To alleviate the first issue that generator representation may contain a big portion of irrelevant information for downstream tasks, we introduce a squeeze (or bottleneck) module $T_\phi$ (Fig. 3) that squeezes informative representations out of the generator representation. In addition, we transform the generated image via a semantic-preserving image transformation $a$ (*e.g.* color jittering and cropping) before feeding it to the student work. Equ. 5 can be rewritten as

$$\min_{\theta,\phi} \; \mathcal{L}_{\mathrm{RD}} = \mathbb{E}_{\mathbf{w} \sim P(\mathbf{w}), a \sim \mathcal{A}} \, \|S_\theta(a[G(\mathbf{w})]) - T_\phi(\mathbf{h}^g(\mathbf{w}))\|_2^2, \qquad (6)$$

where image transformation $a$ is randomly sampled from $\mathcal{A}$. In words, we seek to distill compact representations from the generator among the ones that are invariant to data augmentation $\mathcal{A}$, inspired from the success of recent self-supervised methods [7, 10]. An informal intepretation is that, similar to Chen & He [10], considering one of the alternate subproblems that fix $\theta$ and solve $\phi$, the optimal solution would result in the effect of $T_{\phi^*}(\mathbf{h}^g(\mathbf{w})) \approx \mathbb{E}_{a \sim \mathcal{A}} \, S_\theta(a[G(\mathbf{w})])$, which implies Equ 6 encourages $T_\phi$ to squeeze out transformation-invariant representation. However, similar to the siamese network in SSL [10], there exists a trivial solution to Equ. 6: both the squeeze module and the student network degenerate to output constant for any input.

Therefore, we consult the techniques from SSL methods and add regularization terms to the distillation loss. In particular, we employ variance-covariance [3] to explicitly regularize representations to be

significantly uncorrelated and varied in each dimension. Formally, in a mini-batch of $N$ samples, we denote the squeezed generator representations and student representations with

$$Z_g = [T_\phi(\mathbf{h}^g(\mathbf{w}_1)), T_\phi(\mathbf{h}^g(\mathbf{w}_2)), \ldots, T_\phi(\mathbf{h}^g(\mathbf{w}_N))] \in \mathbb{R}^{M \times N}, \tag{7}$$

$$Z_s = [S_\theta(a_1[G(\mathbf{w}_1)]), S_\theta(a_2[G(\mathbf{w}_2)]), \ldots, S_\theta(a_N[G(\mathbf{w}_N)])] \in \mathbb{R}^{M \times N}, \tag{8}$$

where $\mathbf{w}_i \sim P(\mathbf{w})$ and $a_i \sim \mathcal{A}$ denote random sample of latent variable and data augmentation operator. The variance loss is introduced to encourage the standard deviation of each representation dimension to be greater than 1,

$$\mathcal{L}_{\mathrm{var}}(Z) = \frac{1}{M} \sum_{j=1}^{M} \max(0, 1 - \sqrt{\mathrm{Var}(z^j) + \epsilon}), \tag{9}$$

where $z^j$ represents the $j$-th dimension in representation $\mathbf{z}$. The covariance loss is introduced to encourage the correlation of any pair of dimensions to be uncorrelated,

$$\mathcal{L}_{\mathrm{cov}}(Z) = \frac{1}{M} \sum_{i \neq j} [C(Z)]_{ij}^2,$$
$$\text{where } C(Z) = \frac{1}{N-1} \sum_{i=1}^{N} (\mathbf{z}_i - \bar{\mathbf{z}})(\mathbf{z}_i - \bar{\mathbf{z}})^\top, \ \bar{\mathbf{z}} = \frac{1}{N} \sum_{i=1}^{N} \mathbf{z}_i. \tag{10}$$

To this end, the loss function of squeezing representations from the generator into the student network can be summarized as

$$\mathcal{L}_{\mathrm{squeeze}} = \lambda \mathcal{L}_{\mathrm{RD}} + \mu \left[ \mathcal{L}_{\mathrm{var}}(Z_f) + \mathcal{L}_{\mathrm{var}}(Z_g) \right] + \nu \left[ \mathcal{L}_{\mathrm{cov}}(Z_f) + \mathcal{L}_{\mathrm{cov}}(Z_g) \right]. \tag{11}$$

**Discussion** Our work differs from multi-view representation learning methods [3, 10] in the following aspects. (1) Our work studies the transfer of the generative model that does not originally favor representation extraction, whereas most multi-view representation learning learns representation with discriminative pretext tasks. (2) Unlike typical Siamese networks in multi-view representation learning, the two networks in our work are asymmetric: one takes in noise and outputs an image and the other works in the reverse fashion. (3) While most multi-view representation learning methods learn representation networks from scratch, our work distills representations from a pre-trained model. In specific, most SSL methods create multiview representations by transforming input images in multiple ways, we instead pursue different representation views from a well-trained data generator.

## 4.2 Spanning Representations from Synthetic to Real Domain

Here we address the second problem, the domain between synthetic and real domains, due to two factors. First, the synthesized images may be of low quality. This aspect has been improved a lot with recent GAN modelling [37, 35] and is out of our concern. Second, more importantly, GAN is notorious for the mode collapse issue, suggesting the synthetic data can only cover partial modes of real data distribution. In other words, the synthetic dataset appears to be a subset of the real dataset.

To undermine the harm of mode collapse, we include the real data in the training data of the student network. In particular, in each training step, synthetic data and real data consist of a mini-batch of training data. For synthetic data, the aforementioned squeeze loss is employed. For real data, we employ the original VICReg to compute loss. Specifically, given a mini-batch of real data $\{\mathbf{x}_i^r\}_{i=1}^N$, each image $\mathbf{x}_i^r$ is transformed twice with random data augmentation to obtain two views $a_i(\mathbf{x}_i^r)$ and $a_i'(\mathbf{x}_i^r)$, where $a_i, a_i' \sim \mathcal{A}$. The corresponding representations $Z_r$ and $Z_r'$ are obtained by feeding the transformed images into $S_\theta$ similarly to Equ. 8. Then the loss on real data is computed as

$$\mathcal{L}_{\mathrm{span}} = \lambda \mathcal{L}_{\mathrm{RD}}' + \mu \left[ \mathcal{L}_{\mathrm{var}}(Z_r) + \mathcal{L}_{\mathrm{var}}(Z_r') \right] + \nu \left[ \mathcal{L}_{\mathrm{cov}}(Z_r) + \mathcal{L}_{\mathrm{cov}}(Z_r') \right], \tag{12}$$

where $\mathcal{L}_{\mathrm{RD}}'$ denotes a self-distillation by measuring the distance of two-view representations on real images. The overall loss is computed by simply combine the generated data loss and real data loss as $\mathcal{L}_{\mathrm{total}} = \alpha \mathcal{L}_{\mathrm{squeeze}} + (1 - \alpha)\mathcal{L}_{\mathrm{span}}$, where $\alpha = 0.5$ denotes the proportion of synthetic data in a mini-batch of training samples.

From a technical perspective, spanning representation seems to be a combination of representation distillation and SSL using VICReg [3]. We interpret this combination as spanning representation from the synthetic domain to the real domain. The representation is dominantly learned in the synthetic domain and generalized to the real domain. The student network learns to fuse representation spaces of two domains into a consistent one in the spanning process. Our experimental evaluation shows that "squeeze and span" can outperform VICReg on real data, suggesting that the squeezed representations do have a nontrivial contribution to the learned representation.

# 5 Experiments

## 5.1 Setup

**Dataset and pre-trained GAN**   Our methods are mainly evaluated on CIFAR10, CIFAR100, and STL10, ImageNet100, and ImageNet. **CIFAR10** and **CIFAR100** [39] are two image datasets containing small images at 32×32 resolution with 10 and 100 classes, respectively, and both split into 50,000 images for training and 10,000 for validation. **STL-10** [13], which is derived from the ImageNet [14], includes images at 96×96 resolution over 10 classes. STL-10 contains 500 labeled images per class (*i.e.* 5K in total) with an additional 100K unlabeled images for training and 800 labeled images for testing. **ImageNet100** [55] contains images of 100 classes, among which 126,689 images are regarded as the train split and 5,000 images are taken as the validation split. **ImageNet** [14] is a popular large-scale image dataset of 1000 classes, which is split into 1,281,167 images as training set and 50,000 images as validation set. We adopt StyleGAN2-ADA[5] for representation distillation since it has good stability and high performance. GANs are all pre-trained on training split. More details can be referred in the supplementary material.

**Implementation details**   The squeeze module uses linear layers to transform the generator features into vectors with 2048 dimensions, which are then summed up and fed into a three-layer MLP to get a 2048-d teacher representation. On CIFAR10 and CIFAR100, we use ResNet18 [30] of the CIFAR variant as the backbone. On STL10, we use ResNet18 as the backbone. On ImageNet100 and ImageNet, we use ResNet50 as the backbone. On top of the backbone network, a five-layer MLP is added for producing representation. We use SGD optimizer with cosine learning rate decay [43] scheduler to optimize our models. The actual learning rate is linearly scaled according to the ratio of batch size to 256, *i.e.* $base\_lr \times batch\_size/256$ [24]. We follow the common practice in SSL [7, 55, 29] to evaluate the distilled representation with linear classification task. More details are available in the supplementary material.

## 5.2 Transferring GAN Representation

**Compared methods**   In this section, we justify the advantage of distilling generator representations by comparing the performance of different ways of transferring GAN representation. In particular, we consider the following competitors:

- *Discriminator*. As the discriminator network receives image as input and is ready for representation extraction, we directly extract features, single penultimate features, or multiple features (Equ 2), using a pre-trained discriminator and train a linear classifier on top of them.
- *Encoding*. We train a post hoc encoder with or without real images involved in the training process as in Equ 3.
- *Distilling latent variable*. We employ the vanilla distillation or squeeze method on latent variables with data augmentation engaged.
- *Distilling generator feature*. Our method as described in Section 4.

**Results**   Table 1 presents the comparison results, from which we can draw the following conclusions. (1) Representation distillation, whether from the latent variable or generator feature, significantly outperforms discriminator and encoding. We think this is because image reconstruction and realness discrimination are not suitable pretext tasks for representation learning. (2) Distillation from latent variable achieve comparable performance to distillation from generator feature, despite that the former one show entangled class information (Fig. 2). This result can be attributed to a projection head in the student network. (3) Our method works significantly better than vanilla distillation which does not employ a squeeze module. This result suggests that our method squeeze more informative representation that can help to improve the student performance.

## 5.3 Comparison to SSL

**Linear classification**   We further compare our methods to SSL algorithms such as SimSiam [10] and VICReg [3] in different training data domains: real, synthetic, and a mixture of real and synthetic.

---

[5] https://github.com/NVlabs/stylegan2-ada-pytorch

| Knowledge Source | Transfer Method | Domain | CIFAR10 | CIFAR100 |
|---|---|---|---|---|
| Discriminator | Direct use (single feature) | Syn. & Real | 63.81 | 30.11 |
| | Direct use (multi-feature) | Syn. & Real | 77.58 | 51.63 |
| Latent variable | Encoding | Syn. | 57.15 | 32.19 |
| | Encoding | Syn. & Real | 50.27 | 28.43 |
| | Vanilla distillation (w/ aug) | Syn. | 84.84 | 53.26 |
| | Squeeze | Syn. | 86.99 | 58.56 |
| | Squeeze and span | Syn. & Real | 90.95 | 66.17 |
| Generator feature | Vanilla distillation (w/ aug) | Syn. | 84.48 | 52.77 |
| | Squeeze | Syn. | 87.67 | 57.35 |
| | Squeeze and span | Syn. & Real | 92.54 | 67.87 |

Table 1: **Rrepresentation transfer** from different teachers. Top-1 accuracy of linear classification on CIFAR10 and CIFAR100 validation sets are reported and compared.

| Pretrain Data | Methods | CIFAR10 | CIFAR100 | STL10 | ImageNet100 | ImageNet |
|---|---|---|---|---|---|---|
| Real | SimSiam [10] | 90.94 | 62.44 | 71.30 | – | – |
| | VICReg [3] | 89.20 | 63.31 | 74.43 | – | – |
| Syn | SimSiam [10] | 85.11 | 47.89 | 73.38 | – | – |
| | VICReg [3] | 84.68 | 52.84 | 70.80 | | – |
| | Squeeze (Ours) | 87.67 | 57.35 | 73.35 | – | – |
| Real & Syn | SimSiam [10] | 90.88 | 62.68 | 71.70 | – | – |
| | VICReg [3] | 90.46 | 65.22 | 75.05 | 46.42 | 47.32 |
| | Sq & Sp (Ours) | **92.54** | **67.87** | **76.83** | **53.32** | **47.80** |

Table 2: **Linear classification performance comparison** to seminal SSL methods. Top-1 accuracy on validation set is reported. The biggest number is **bolded** and the second biggest number is underlined.

Table 2 presents the linear classification results, from which we want to highlight the following points. (1) Both SimSiam and VICReg perform worse when pre-trained on only synthetic data than only real data, indicating the existence of a domain gap between synthetic data and real data. (2) Our methods outperform SimSiam and VICReg in synthetic and mixture domains, suggesting distillation of generator feature contributes extra improvement SSL. (3) Our "Squeeze and Span" is the best among all the competitors on CIFAR10, CIFAR100, and STL10. (4) Our method outperforms VICReg with a large margin (6.90% Top-1 Acc) on ImageNet100 and a clear increase (0.48% Top-1 Acc) on ImageNet.

**Transfer learning** As one goal of representation learning is its transferability to other datasets, we further conduct a comprehensive transfer learning evaluation. We follow the protocol in [20] and use its released source code[6] to conduct a thorough transfer learning evaluation for our pre-trained models on ImageNet100/ImageNet. In particular, the learned representations are mainly evaluated for (1) linear classification on 11 datasets including Aircraft, Caltech101, Cars, CIFAR10, CIFAR100, DTD, Flowers, Food, Pets, SUN397, and VOC2007; (2) finetuning on three downstream tasks and datasets, including object detection on PASCAL VOC, surface normal estimation on NYUv2, and semantic segmentation on ADEChallenge2016. Please refer to [20] for the details of evaluation protocol.

The results are presented in Table 3 and Table 4, where we have the following observation (1) As depicted in Table 3, our method achieves better transferability than VICReg on the mixed data no matter pre-trained on ImageNet100 or ImageNet. Our method beats VICReg on nearly all other datasets and the improvement on average accuracy is 3.40 with models pre-trained on ImageNet100 and 1.00 with models pre-trained on ImageNet. (2) As depicted in Table 4, representations learned with our method can be well transferred to various downstream tasks such as object detection,

---

[6]https://github.com/linusericsson/ssl-transfer

| Pre-training Data | Method | Aircraft | Caltech101 | Cars | CIFAR10 | CIFAR100 | DTD | Flowers | Food | Pets | SUN397 | VOC2007 | Avg. |
|---|---|---|---|---|---|---|---|---|---|---|---|---|---|
| ImageNet100 | VICReg | **23.96** | 60.29 | 15.27 | 80.28 | 57.11 | 45.95 | 60.26 | 33.40 | 38.41 | 29.53 | 49.07 | 44.86 |
| (Syn.&Real) | Sq&Sp (Ours) | 23.88 | **63.59** | **15.30** | **84.37** | **61.28** | **49.36** | **63.41** | **37.80** | **43.28** | **33.04** | **55.64** | **48.26** |
| ImageNet | VICReg | 32.39 | 79.49 | 22.37 | 90.09 | 70.67 | 58.88 | 79.97 | 50.55 | 59.47 | 45.76 | **68.74** | 59.85 |
| (Syn.&Real) | Sq&Sp (Ours) | **33.85** | **80.65** | **25.71** | **90.14** | **70.81** | **61.75** | **80.09** | **50.84** | **61.01** | **46.34** | 68.30 | **60.86** |

Table 3: Linear classification performance on 11 downstream classification datasets.

| Pre-training Data | Method | VOC Detection | | | NYUv2 Surface Normal Estimation | | | | | ADE Semantic Segmentation | |
|---|---|---|---|---|---|---|---|---|---|---|---|
| | | AP ↑ | AP50 ↑ | AP75 ↑ | Mean ↓ | Median ↓ | 11.25° ↑ | 22.5° ↑ | 30° ↑ | Mean IoU ↑ | Accuracy ↑ |
| ImageNet100 | VICReg | 33.75 | 61.73 | 31.77 | 34.62 | 29.70 | 20.90 | 39.77 | 50.40 | **0.3008** | 73.19 |
| (Syn.&Real) | Sq&Sp (Ours) | **41.10** | **69.07** | **42.54** | **33.09** | **27.47** | **22.90** | **42.71** | **53.47** | 0.2993 | **73.25** |
| ImageNet | VICReg | 46.69 | 75.78 | 49.07 | 33.86 | 28.42 | 22.08 | 41.35 | 52.15 | 0.3263 | 74.85 |
| (Syn.&Real) | Sq&Sp (Ours) | **48.98** | **77.50** | **52.85** | 33.39 | 28.30 | 22.27 | 41.65 | 52.26 | **0.3299** | **75.22** |

Table 4: Downstream task finetuning performance, including object detection on PASCAL VOC, surface normal estimation on NYUv2, and semantic segmentation on ADEChallenge2016. ↑ denotes higher is better while ↓ denotes lower is better.

surface normal estimation and semantic segmentation, and consistently show higher performance than VICReg.

We believe these results suggest that generator features have strong transferability and great promise to contribute to self-supervised representation learning.

## 5.4 Ablation Study

**Effect of squeeze and span** The effect of our method is studied by adding modules to the vanilla version of representation distillation (a) one by one. (a) → (b): After added data augmentation, significant improvement can be observed, suggesting that invariant representation to data augmentation is crucial for linear classification performance. This result inspires us to make teacher representation more invariant. (b) → (c): the learnable $T_\phi$ is introduced to squeeze out invariant representation as teacher. However, trivial performance (10% top-1 accuracy, no better than random guess) is obtained, implying models learn trivial solutions, probably constant output. (c) → (d) & (e): regularization terms are added, and the student network now achieves meaningful performance, which indicates the trivial solution is prevented. Moreover, using both regularizations achieves the best performance, which outperforms (b) without "squeeze". (e) → (f): training data is supplemented with real data, *i.e.* adding "span", the performance is further improved.

**Domain gap issue** We calculate the squared MMD [4] of representations of synthetic and real data to measure their gap in representation space. Table 6 shows "Squeeze and Span" (Sq&Sp) reducesthe MMD compared to "Squeeze" by an order of magnitude on CIFAR10 and CIFAR100 and a large margin on STL10, clearly justifying the efficacy of "span" as reducing the domain gap.

**Impact of generator** We further compare the performance of our method when we use GAN checkpoints of different quality. Fig. 4 shows the top-1 accuracy with respect to FID, which indicates the quality of GAN. It is not surprising that GAN quality significantly impacts our method. The higher the quality of generator we utilize, the higher performance of learned representation our method can attain. It is noteworthy that a moderately trained GAN (FID < 11.03) is already able to contribute additional performance improvement on CIFAR100 when compared to VICReg trained on a mixture of synthetic and real data. In the appendix, we further analyze the im-

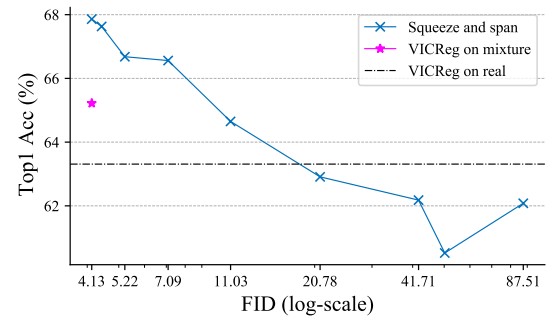

Figure 4: Representation performance (top-1 accuracy) versus generator quality (FID) on CIFAR100. A better GAN has a lower FID.

| | $\mathcal{L}_{\text{RD}}$ | $\mathcal{A}$ | $T_\phi$ | $\mathcal{L}_{\text{var}}$ | $\mathcal{L}_{\text{cov}}$ | Span | Top-1 Acc |
|---|---|---|---|---|---|---|---|
| a | ✓ | | | | | | 74.20 |
| b | ✓ | ✓ | | | | | 84.48 |
| c | ✓ | ✓ | ✓ | | | | 10.00 |
| d | ✓ | ✓ | ✓ | ✓ | | | 79.10 |
| e | ✓ | ✓ | ✓ | ✓ | ✓ | | 87.67 |
| f | ✓ | ✓ | ✓ | ✓ | ✓ | ✓ | 92.54 |

Table 5: **Ablation study** on CIFAR10. $T_\phi$ and $\mathcal{A}$ denote whether to introduce the learnable squeeze module and data augmentation, respectively. $\mathcal{L}_{\text{RD}}$, $\mathcal{L}_{\text{var}}$, and $\mathcal{L}_{\text{cov}}$ represent whether to enable the correpsonding losses.

| Methods | Pretrain Data | CIFAR10 ($\times 10^{-5}$) | CIFAR10- ($\times 10^{-5}$) | STL10 ($\times 10^{-3}$) |
|---|---|---|---|---|
| VICReg [3] | Syn | 3.44 | 5.89 | 5.39 |
| | Real & Syn | 3.74 | 16.8 | 11.4 |
| Squeeze (Ours) | Syn | 4.79 | 1.24 | 9.82 |
| Sq & Sp (Ours) | Real & Syn | 0.45 | 0.25 | 3.71 |

Table 6: Squared MMD between synthetic and real data representation that measures the discrepancy of representation across domains. Lower number indicates smaller domain gap.

pact of generator feature choices and GAN architectures on the distillation performance.

# 6 Conclusions

This paper proposes to "squeeze and span" representation from the GAN generator to extract transferable representation for downstream tasks like image classification. The key techniques, "squeeze" and "span", aim to mitigate issues that the GAN generator contains the information necessary for image synthesis but unnecessary for downstream tasks and the domain gap between synthetic and real data. Experimental results justify the effectiveness of our method and show its great promise in self-supervised representation learning. We hope more attention can be drawn to studying GAN for representation learning.

**Limitation and future work** The current form of our work still maintains several limitations that need to be studied in the future. (1) Since we distill representation from GANs, the performance of learned representation relies on the quality of pretrained GANs and thus is limited by the performance of the GAN techniques. Therefore, whether a prematurely trained GAN can also contribute to self-supervised representation learning and how to effectively distill them will be an interesting problem. (2) In this paper, the squeeze module sets the widely-used transformation-invariance as the learning objective of representation distillation. We leave other learning objectives tailored for specific downstream tasks as future work. (3) More comprehensive empirical study with larger scale is left as future work to further exhibit the potential of our method.

# Acknowledgments and Disclosure of Funding

This work was supported by the National Key R&D Program of China under Grant 2018AAA0102801, National Natural Science Foundation of China under Grant 61620106005, EPSRC Visual AI grant EP/T028572/1.

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
