# Distilling Representations from GAN Generator via Squeeze and Span: Appendix

## A   Implementation Details

**Pre-trained GAN**   Table 1 presents the details of pre-trained GANs employed in our experiments. The GAN checkpoints with the lowest FID are selected.

| Dataset | Resolution | FID | Source |
|---|---|---|---|
| CIFAR10 | $32 \times 32$ | 2.92 | Publicly available at cifar10u-cifar-ada-best-fid.pkl |
| CIFAR100 | $32 \times 32$ | 4.13 | Trained with StyleGAN2-ADA [4] by ourselves using configuration: `--cfg=cifar` |
| STL10 | $128 \times 128$ | 20.86 | Trained with StyleGAN2-ADA [4] by ourselves using configuration: `kimg=25000, mb=64, mbstd=8, fmaps=0.5, lrate=0.0025, gamma=0.25, map=8` |
| ImageNet100 | $128 \times 128$ | 23.7 | Trained with Projected GAN [6] with StyleGAN2 architecture using `cfg=stylegan2, batch=128, mirror=1, kimg=25000` |
| ImageNet | $128 \times 128$ | 49.8 | StyleGAN2 publicly available at gan-transfer repo |

Table 1: Pretrained GANs.

**Default representation source**   Although Table 1 in the main text shows that good performance can be obtained by distillation from both latent variable and generator feature, we choose generator feature (Equ 4 in the main text) as default since generator feature shows better properties (Fig. 2 in the main text) than others. Moreover, this choice can easily apply to other GAN architecture like BigGAN [2].

**Hyperparameters**   Table 2 presents the hyperparameters for distilling GAN for different dataset.

| | Aug | Batch size | Epochs | `base_lr` | Weight decay | Scale | $\lambda$ | $\mu$ | $\nu$ | $\alpha$ |
|---|---|---|---|---|---|---|---|---|---|---|
| CIFAR10 | MoCo-v2 aug w/o blur | 512 | 800 | 0.03 | 0.0005 | 32 | 25 | 25 | 1 | 0.5 |
| CIFAR100 | MoCo-v2 aug w/o blur | 512 | 800 | 0.03 | 0.0005 | 32 | 10 | 10 | 1 | 0.5 |
| STL10 | MoCo-v2 aug | 512 | 200 | 0.05 | 0.0001 | 64 | 25 | 25 | 1 | 0.5 |
| ImageNet100 | MoCo-v2 aug | 256 | 100 | 0.05 | 0.0001 | 96 | 25 | 25 | 1 | 0.5 |
| ImageNet | MoCo-v2 aug | 256 | 100 | 0.05 | 0.0001 | 96 | 25 | 25 | 1 | 0.5 |

Table 2: Hyperparameters for distilling GAN representations on CIFAR10, CIFAR100, STL10, ImageNet100, and ImageNet.

Furthermore, Table 3 and Table 4 present the hyperparameters for training VICReg and SimSiam, respectively.

| | Aug | Batch size | Epochs | base_lr | Weight decay | Scale | $\lambda$ | $\mu$ | $\nu$ | $\alpha$ |
|---|---|---|---|---|---|---|---|---|---|---|
| CIFAR10 | MoCo-v2 aug w/o blur | 512 | 800 | 0.03 | 0.0005 | 32 | 25 | 25 | 1 | 0.5 |
| CIFAR100 | MoCo-v2 aug w/o blur | 512 | 800 | 0.03 | 0.0005 | 32 | 25 | 25 | 1 | 0.5 |
| STL10 | MoCo-v2 aug | 512 | 200 | 0.05 | 0.0001 | 64 | 25 | 25 | 1 | 0.5 |
| ImageNet100 | MoCo-v2 aug | 256 | 100 | 0.05 | 0.0001 | 96 | 25 | 25 | 1 | 0.5 |
| ImageNet | MoCo-v2 aug | 256 | 100 | 0.05 | 0.0001 | 96 | 25 | 25 | 1 | 0.5 |

Table 3: Hyperparameters for training VICReg [1] on CIFAR10, CIFAR100, STL10, ImageNet100, and ImageNet.

| | Aug | Batch size | Epochs | base_lr | Weight decay | Scale | $\alpha$ |
|---|---|---|---|---|---|---|---|
| CIFAR10 | MoCo-v2 aug w/o blur | 512 | 800 | 0.03 | 0.0005 | 32 | 0.5 |
| CIFAR100 | MoCo-v2 aug w/o blur | 512 | 800 | 0.03 | 0.0005 | 32 | 0.5 |
| STL10 | MoCo-v2 aug | 512 | 200 | 0.05 | 0.0001 | 64 | 0.5 |

Table 4: Hyperparameters for training SimSiam [3] on CIFAR10, CIFAR100, and STL10.

# B  Further Analysis

## B.1  CIFAR and STL cross-dataset evaluation

We train the feature extractor on one of CIFAR10, CIFAR100, and STL10 datasets and run linear classification on the rest two datasets. Results in Table 5 show that our method (Sq & Sp) generally achieves better cross-dataset generalization compared to SimSiam and VICReg.

| Pretrain Data | Methods | CIFAR10 | | CIFAR100 | | STL10 | |
|---|---|---|---|---|---|---|---|
| | | CIFAR100 | STL10 | CIFAR10 | STL10 | CIFAR10 | CIFAR100 |
| | SimSiam [3] | 41.32 | 68.70 | 75.90 | 60.41 | 59.34 | 32.88 |
| Real & Syn | VICReg [1] | 54.19 | 80.56 | 79.11 | **74.73** | 58.15 | 31.25 |
| | Sq & Sp (Ours) | **58.93** | **82.22** | **80.68** | 73.85 | **64.76** | **37.48** |

Table 5: Cross-dataset linear classification performance. The first and second rows show the source and target dataset respectively.

## B.2  Generator Feature Choices

To shed light on the property of generator features, experiments over different generator feature choices are conducted. In particular, we consider feature maps from generator blocks at different resolution, e.g. "b16" represents the synthesis block at $16 \times 16$ resolution. Feature maps are grouped into three levels of scale: b4-b8 (small), b16-b32 (middle), and b64-b128 (large). Results of an ablation study with respect to generator features on "squeeze" method across CIFAR10, CIFAR100, and STL10 are presented in Table 6. It shows that using lower-resolution (e.g. b4 – b8) feature maps leads to slightly better performance than higher-resolution (e.g. b16 – 32 or b64 – b128). This conclusion is in accordance with common understanding of network features that low-resolution feature is more abstract than high-resolution feature and benificial for high-level discriminative tasks like classification. The strategy of using feature maps from all resolutions yields superior performance on CIFAR10 and STL10 and comparable performance with b4-b8 on CIFAR100.

Furthermore, we provide an in-depth analysis of the cross-layer generator feature similarity. The globally average pooled features from different convolution layers in the geneartor are computed the pairwise CKA [5] similarity, shown as the confusion matrix in Fig. 1, where we also present results of ResNet18 of CIFAR variant learned by SimSiam for comparison. It is interesting to see that in generator most similarity is lower than 0.8, showing unique and complementary features across different layers. On the contrary, in ResNet18 many pairs of features have similarity greater than 0.85, suggesting duplicate features. These results support our motivation to distill generator features and choose all block feature maps.

| Pretrain Data | Methods | Generator Block | CIFAR10 | CIFAR100 | STL10 |
|---|---|---|---|---|---|
| Syn | Squeeze | b4–b8 | 87.05 | **57.51** | 74.05 |
| | | b16–b32 | 87.08 | 55.43 | 73.43 |
| | | b64–b128 | – | – | 73.08 |
| | | All | **87.67** | 57.35 | **76.83** |

Table 6: Ablation with respect to generator blocks.

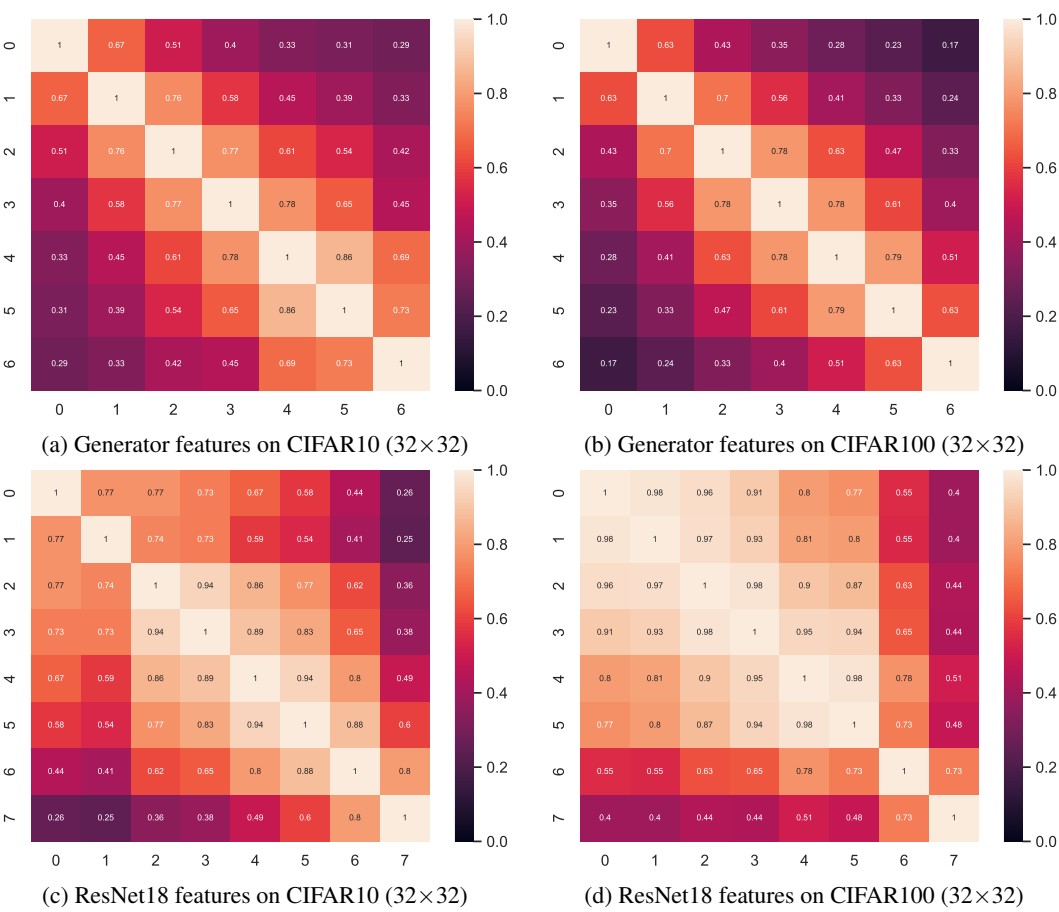

(a) Generator features on CIFAR10 (32×32)

(b) Generator features on CIFAR100 (32×32)

(c) ResNet18 features on CIFAR10 (32×32)

(d) ResNet18 features on CIFAR100 (32×32)

Figure 1: CKA similarity of features across different network layers. Generator features and ResNet18 of CIFAR variant features learned by SimSiam on CIFAR10 and CIFAR100 are compared.

## B.3 Different GAN Architectures

As different architecture may have different inductive bias, we further conduct a study on the impact of architecture on the distillation performance. We train and evaluate our "squeeze" method with the generator features that are obtained from different GAN architectures including AutoGAN, StyleGAN-XL, and BigGAN on CIFAR10, and report the results in Table 7. We use the same feature selection strategy across different architectures as before: the consecutive network layers are grouped into a block according to its output resolution and the output feature maps of block at each resolution are chosen.

Table 7 presents whether the GAN is pretrained in conditional or unconditional manner, the link to publicly available pre-trained checkpoints, their FID, and the performance of "squeeze" method. From these results, we would like to highlight the following points: (1) Our method yields good

performance with prevalent GAN architectures such as StyleGAN2, StyleGAN3, and BigGAN, showing that our method is architecture-agnostic. (2) When the generators are conditioned on class labels, they result in better distillation and hence classification performance than unconditional ones, possibly due to its higher generation quality (conditional StyleGAN2-ADA achieves 0.5 FID lower than unconditional one) and the embedded class information in conditional modeling. Since training conditional GAN requires class labels which violates the goal of unsupervised learning, we only report these results for sake of completeness. (3) Although StyleGAN-XL with StyleGAN3 architecture achieves the highest generation quality (1.85 FID), its distillation performance is 2.70 top-1 acc lower than StyleGAN2. This suggests that generation quality may not be the only factor determining the representation transfer ability. We hope to further study this problem in the future.

| GAN architectures | Type | Sources | Generation FID | Squeeze Top-1 Acc |
|---|---|---|---|---|
| StyleGAN2-ADA | Unconditional | GitHub repo, model | 2.92 | 87.67 |
| AutoGAN | Unconditional | GitHub repo, model | 12.42 | 76.28 |
| StyleGAN2-ADA | Conditional | GitHub repo, model | 2.42 | **88.90** |
| StyleGAN-XL (StyleGAN3) | Conditional | GitHub repo, model | **1.85** | 84.97 |
| BigGAN-DiffAugment-cr | Conditional | GitHub repo, model | 8.49 | 86.41 |

Table 7: Ablation study with respect to GAN architecture. The best results are in bold and the second best is underlined.