# OpenReview forum: "Distilling Representations from GAN Generator via Squeeze and Span"
_NeurIPS.cc/2022/Conference — NeurIPS 2022 Accept_

### Official Review · Reviewer_8CJZ · 2022-07-10

**Rating:** 5
**Confidence:** 4
**Soundness:** 3 good
**Presentation:** 3 good
**Contribution:** 2 fair

**Summary:**

The paper proposes a method to distill representations from a pretrained GAN generator for the downstream classification tasks. The paper further proposes a "squeeze" technique to make the representation compact, and a "span" technique to mitigate the synthetic to real domain gap. Experimental results on small datasets including CIFAR10, CIFAR100, and STL10 show its efficiency in self-supervised representation learning.

**Questions:**

See "Weaknesses".

**Limitations:**

Yes.

**Strengths And Weaknesses:**

__Strengths__

- This paper is well-written and easy to follow.
- The idea of distilling representations from GAN Generators is well-motivated.
- The proposed squeeze and span techniques mitigate the mentioned problems effectively.

__Weaknesses__

- The proposed squeeze and span techniques are heavily inspired by VICReg, which downweights the novelty of the paper.
- From a technical perspective, distillation from "latent variable" and "generator feature" are not that very different. This can be verified by the numbers in Table 1, the vanilla distillation and squeeze accuracies are very similar. I also wonder what are accuracies of the "squeeze and span" for "latent variable" and how would those compare to the proposed full model.
- It is hard to assess the effectiveness of the proposed method without experiments on ImageNet for such an empirical paper. I understand that the limited computational resources might be a concern, but any form of ImageNet like low resolution of 64x64 or TinyImageNet (which has 200 classes) would be helpful.

__Post Rebuttal__

I thank authors for their response. Some of my concerns have been addressed. But my concern on missing results on large-scale datasets still remain. Therefore, I keep my initial rating but lean towards acceptance for this paper.

---

> ### Author Response · Authors · 2022-08-02
> **Response to Reviewer 8CJZ**
>
> **1. The proposed squeeze and span techniques are heavily inspired by VICReg, which downweights the novelty of the paper.**
>
> Thanks for your comment. Applying SSL technique is a common way in representation distillation literature [1]. In our work, the VICReg serves as a technique to instantiate our “squeeze and span” framework for distilling the generator features. We think the novelty of our paper mainly lies in proposing a simple yet effective framework to transfer generator features, which is different from existing methods either using discriminator or encoder (Fig. 1 in the manuscript).
>
> **2. What are accuracies of the "squeeze and span" for "latent variable" and how would those compare to the proposed full model.**
>
>
> Thanks for your interest. We have run the suggested experiments on CIFAR10 and CIFAR100 and compare the results with the proposed full model, shown in the following table.
>
> | Knowledge Source | Transfer Method | Domain | CIFAR10 | CIFAR100 |
> | :---: | :---: | :---: | :---: | :---: |
> | Latent variable | Squeeze and span | Syn. & Real | 90.95 | 66.17 |
> | Generator feature | Squeeze and span | Syn. & Real | 92.54 (+ 1.59) | 67.87 (+ 1.70) |
>
> It shows that applying the squeeze and span on generator features outperforms latent variables. distill generator features instead of latent variable by default out of two consideration:
> Generator feature leads to better empirical performance.
> Distilling generator feature is more applicable to various generator architectures. Most generator achitectures have intermediate feature maps while not all the generator architectures have properly learned latent space like StyleGAN.
>
> **3. Any form of ImageNet like low resolution of 64x64 or TinyImageNet (which has 200 classes) would be helpful.**
>
> We have run a small scale version of ImageNet experiment. We first train a Projected GAN [2] with StyleGAN2 architecture on ImageNet100 [3] at 128x128 resolution until 25M images have been shown to the discriminator and obtain a pre-trained generator with 23.78 FID. Second, we apply “squeeze and span” to distill the generator features and run VICReg on the mixture of real and synthetic data for comparison. The linear classification performance of learned representation on ImageNet100 validation set is evaluated. The configuration and results are listed as below. Our method outperforms VICReg on mixed data by a large margin (6.90% Top-1 Acc),showing the great promise of generator representation in SSL.
>
> | GAN | Methods | Network | Input Resolution | Batch size | Epoch | Top1 Acc | Top5 Acc |
> | :---: | :---: | :---: | :---: | :---: | :---: | :---: | :---: |
> | Projected GAN | Squeeze and Span (Ours) | ResNet50 | 96 | 256 | 100 | 53.32 | 81.04 |
> | Projected GAN | VICReg on mixed data | ResNet50 | 96 | 256 | 100 | 46.42 | 76.06 |
>
>
> **Reference**
> - [1] Yonglong Tian, Dilip Krishnan, Phillip Isola. Contrastive Representation Distillation. In ICLR, 2020.
> - [2] Axel Sauer, Kashyap Chitta, Jens Müller, Andreas Geiger. Projected GANs Converge Faster. In NeurIPS, 2021.
> - [3] Yonglong Tian, Dilip Krishnan, and Phillip Isola. Contrastive multiview coding. In ECCV, 2020.

---

> > ### Author Response · Authors · 2022-08-09
> > **Additional ImageNet results**
> >
> > Please check the further response to reviewer pYVQ and our revised submission for additional ImageNet results. We hope these results could address your concern.

---

### Official Review · Reviewer_pYVQ · 2022-07-11

**Rating:** 6
**Confidence:** 5
**Soundness:** 2 fair
**Presentation:** 3 good
**Contribution:** 2 fair

**Summary:**

This paper proposes to distill representations from GANs for downstream tasks. Specifically, it emphasizes the importance of generator features and proposes a squeeze and span solution to effectively extract such generator features as image representations. To do so, it learns a feature prediction network that predicts generator features transformed under some kind of semantic-preserving transformations. So that generator features of real images can be directly computed. Moreover, to reduce the gap between real and fake images and avoid the generator representation of a real image suffer from GANs' issues such as mode collapse, the proposed method further applies a spanning operation that trains the feature prediction network also on real images using VICReg.

**Questions:**

Overall I think current experiments are significantly insufficient to comprehensively study the effectiveness and different properties of GAN generator features as self-supervised representations. My detailed concerns are covered in weaknesses. I believe studying generator features as self-supervised representations across generator architectures, feature depths, downstream tasks, as well as test datasets will lead to many more insightful observations.

**Limitations:**

Authors discussed the limitations in the main manuscript, which look reasonable to me.

**Strengths And Weaknesses:**

Strengths:
+ The task of distilling GAN features for downstream perception tasks is definitely a worthexploring direction

+ The manuscript is easy to follow

+ The visualization included in Figure 2 well motivates the use of generator features over latent variables and discriminator features.

Weaknesses:
- one missing related work:

[a] Generative Hierarchical Features from Synthesizing Images, CVPR 2021.

which also utilizes generator features for various downstream tasks.

- Experiments should be significantly extended. Specifically, authors only conduct experiments on low-resolution images of CIFAR10, CIFAR100 and SHVN. As pointed out by the authors as the third point of the limitations, the proposed method should be validated on more challenging datasets with higher resolution images. I believe this is a must. Since practical value is an important aspect for self-supervised learning methods, CIFAR10, CIFAR100 and SHVN are insufficient to reflect the richness and informativeness of generator features.

- Authors only used StyleGAN2-ADA as the target generator to distill from, which is not sufficient. Studying the generator features of different GAN architectures is important for studying the effectiveness of generator features since different architectures naturally have different inductive bias.

- Authors only used distilled generator features on the test set of the same dataset the gan is trained on. However, one property of self-supervised methods is that their representations are generally useful across datasets. Authors should include such a study.

- Authors only used distilled generator features on image classification. Other downstream tasks are needed to study the pros&cons of generator features as a self-supervised representation. In fact, maybe authors should follow the standard protocol for evaluating self-supervised methods.

- Just concatenate all layer features as the representation provided by the generator is insufficient to obtain solid observations. Authors should include studies that use different features or combinations from the single generator.

---

> ### Author Response · Authors · 2022-08-02
> **Response to Reviewer pYVQ [2/2]**
>
> **5. Other downstream tasks are needed to study the pros&cons of generator features as a self-supervised representation.**
>
> We are going to run downstream task evaluation such as object detection and instance segmentation on PASCAL VOC as in [7] for our trained models on ImageNet-like datasets. We hope to include these additional results in the discussion period.
>
> **6.Authors should include studies that use different features or combinations from the single generator.**
>
> As suggested, we have run experiments over different generator feature choices. In particular, we use feature maps from generator blocks at different resolutions, e.g. b16 represents the synthesis block at 16x16 resolution. For simplicity and without lossing the generality, we consider three groups of feature maps: b4-b8, b16-b32, and b64-b128. Ablation study is done on “squeeze” method across CIFAR10, CIFAR100, and STL10. The results listed below show that using lower-resolution (e.g. b4-b8) feature maps leads to slightly better performance than higher-resolution (e.g. b16-32 or b64-b128). This conclusion is in accordance with common understanding of network features that low-resolution feature is more abstract than high-resolution feature and benificial for high-level discriminative tasks like classification. The best strategy is still using feature maps from all resolutions.
>
> | Pretrain Data | Method | Generator block | CIFAR10 | CIFAR100 | STL10 |
> | :---: | :---: | :---: | :---: | :---: | :---: |
> | Syn | Squeeze | b4-b8 | 87.05 | 57.51 | 74.05 |
> | Syn | Squeeze | b16-b32 | 87.08 | 55.43 | 73.43|
> | Syn | Squeeze | b64-b128 | --- | --- | 73.08 |
> | Syn | Squeeze | All | 87.67 | 57.35 | 76.83 |
>
> Furthermore, we provide an in-depth analysis of the cross-layer generator feature similarity. The globally average pooled features from different convolution layers in the geneartor are computed the pairwise CKA [8] similarity, shown as the confusion matrix in the Fig.1 of appendix, where we also present results of ResNet18 of CIFAR variant learned by SimSiam for comparison. It is interesting to see that in generator most similarity is lower than 0.8, showing unique and complementary features across different layers.
> On the contrary, in ResNet18 many pairs of features have similarity greater than 0.85, suggesting duplicate features.
> These results support our motivation to distill generator features and choose all block feature maps.
>
>
> **References**
> - [1] Yinghao Xu, Yujun Shen, Jiapeng Zhu, Ceyuan Yang, and Bolei Zhou. Generative hierarchical features from synthesizing images. In CVPR, 2021.
> - [2] Tianyu Hua, Wenxiao Wang, Zihui Xue, Sucheng Ren, Yue Wang, and Hang Zhao. On Feature Decorrelation in Self-Supervised Learning. In CVPR, 2021.
> - [3] Zhiqiang Shen, Zechun Liu, Zhuang Liu, Marios Savvides, Trevor Darrell, and Eric Xing. Un-Mix: Rethinking Image Mixtures for unsupervised Visual Representation Learning. In AAAI, 2022.
> - [4] Xinyu Gong, Shiyu Chang, Yifan Jiang, Zhangyang Wang. AutoGAN: Neural Architecture Search for Generative Adversarial Networks. In ICCV, 2019.
> - [5] Tero Karras, Miika Aittala, Samuli Laine, Erik Härkönen, Janne Hellsten, Jaakko Lehtinen, and Timo Aila. Alias-free generative adversarial networks. NeurIPS, 2021.
> - [6] Andrew Brock, Jeff Donahue, and Karen Simonyan. Large scale gan training for high fidelity natural image synthesis. In ICLR, 2018.
> - [7] Kaiming He, Haoqi Fan, Yuxin Wu, Saining Xie, and Ross Girshick. Momentum contrast for unsupervised visual representation learning. In CVPR, 2020.
> - [8] Simon Kornblith, Mohammad Norouzi, Honglak Lee, Geoffrey E. Hinton. Similarity of Neural Network Representations Revisited. In ICML, 2019.

---

> > ### Comment · Reviewer_pYVQ · 2022-08-07
> > **Response to authors' response**
> >
> > I appreciate authors feedback with extended evaluation of their method as suggested. While waiting for the results of the additional experiments mentioned by the authors, I believe the additional results on cross-dataset evaluation, ImageNet, and different feature choices have significantly improved the quality of this submission.

---

> > > ### Author Response · Authors · 2022-08-09
> > > **Further Response to Reviewer pYVQ [3/3]**
> > >
> > > | Dataset | Methods | Top1 Acc | Top5 Acc |
> > > | :---: | :---: | :---: | :---: |
> > > | ImageNet100 | VICReg on mixed data | 46.42 | 76.06 |
> > > | ImageNet100 | Squeeze and Span (Ours) | **53.32** | **81.04** |
> > > | ImageNet | VICReg on mixed data | 47.32 | 71.98 |
> > > | ImageNet | Squeeze and Span (Ours) | **47.80** | **72.45** |
> > >
> > > Table 2-1: Results of linear classification evaluation on the test set of the same dataset. The best results in each dataset is bolded.
> > >
> > >
> > > | Pre-training Data | Method | Aircraft | Caltech101 | Cars | CIFAR10 | CIFAR100 | DTD | Flowers | Food | Pets | SUN397 | VOC2007 | Avg. |
> > > | :---: | :---: | :---: | :---: | :---: | :---: | :---: | :---: |:---: | :---: | :---: | :---: | :---: | :---: |
> > > | IN100 | VICR | **23.96** | 60.29 | 15.27 | 80.28 | 57.11 | 45.95 | 60.26 | 33.40 | 38.41 | 29.53 | 49.07 | 44.86 |
> > > | IN100 | Sq&Sp | 23.88 | **63.59** | **15.30** | **84.37** | **61.28** | **49.36** | **63.41** | **37.80** | **43.28** | **33.04** | **55.64** | **48.26** |
> > > | IN | VICR | 32.39 | 79.49 | 22.37 | 90.09 | 70.67 | 58.88 | 79.97 | 50.55 | 59.47 | 45.76 | **68.74** | 59.85 |
> > > | IN | Sq&Sp | **33.85** | **80.65** | **25.71** | **90.14** | **70.81** | **61.75** | **80.09** | **50.84** | **61.01** | **46.34** | 68.30 | **60.86** |
> > >
> > > Table 2-2: Results of linear classification evaluation on other 11 datasets. IN and IN100 are short for ImageNet and ImageNet100. VICR represents VICReg trained on a mixture of real and synthetic data. Sq&Sp represents our “squeeze and span” method. The best results in each dataset is bolded.
> > >
> > >
> > > |  |  | VOC | Detection | (finetune) | NYUv2 | Surface  | Normal  | Estimation | | ADE Semantic | Segmentation |
> > > | :---: | :---: | :---: | :---: | :---: | :---: | :---: | :---: |:---: | :---: | :---: | :---: |
> > > | Pre-training Data | Method | AP ↑ | AP50 ↑ | AP75 ↑ | Mean ↓ | Median ↓ | 11.25° ↑ | 22.5° ↑ | 30° ↑ | Mean IoU ↑ | Accuracy ↑ |
> > > | IN100 | VICR | 33.75 | 61.73 | 31.77 | 34.62 | 29.70 | 20.90 | 39.77 | 50.40 | **0.3008** | 73.19 |
> > > | IN100 | Sq&Sp | **41.10** | **69.07** | **42.54** | **33.09** | **27.47** | **22.90** | **42.71** | **53.47** | 0.2993 | **73.25** |
> > > | IN | VICR | 46.69 | 75.78 | 49.07 | 33.86 | 28.42 | 22.08 | 41.35 | 52.15 | 0.3263 | 74.85 |
> > > | IN | Sq&Sp | **48.98** | **77.50** | **52.85** | **33.39** | **28.30** | **22.27** | **41.65** |**52.26** | **0.3299** | **75.22** |
> > >
> > > Table 2-3: Results of downstream task finetuning evaluation on other datasets, including object detection on PASCAL VOC, surface normal estimation on NYUv2, and semantic segmentation on ADEChallenge2016. IN and IN100 are short for ImageNet and ImageNet100, respectively. VICR represents VICReg trained on a mixture of real and synthetic data. Sq&Sp represents our “squeeze and span” method. ↑ denotes higher is better while ↓ denotes lower is better. The best results in each dataset is bolded.
> > >
> > > **Reference**
> > > - [1] Xinyu Gong, Shiyu Chang, Yifan Jiang, Zhangyang Wang. AutoGAN: Neural Architecture Search for Generative Adversarial Networks. In ICCV, 2019.
> > > - [2] Axel Sauer, Katja Schwarz, and Andreas Geiger. StyleGAN-XL: Scaling StyleGAN to Large Diverse Datasets. In SIGGRAPH, 2022.
> > > - [3] Andrew Brock, Jeff Donahue, and Karen Simonyan. Large scale gan training for high fidelity natural image synthesis. In ICLR, 2018.
> > > - [4] Axel Sauer, Kashyap Chitta, Jens Müller, Andreas Geiger. Projected GANs Converge Faster. In NeurIPS, 2021.
> > > - [5] Timofey Grigoryev, Andrey Voynov, Artem Babenko. When, Why, and Which Pretrained GANs Are Useful? In ICLR, 2022.
> > > - [6] Linus Ericsson, Henry Gouk, Timothy M. Hospedales. How Well Do Self-Supervised Models Transfer? In CVPR, 2021.

---

> > > ### Author Response · Authors · 2022-08-09
> > > **Further Response to Reviewer pYVQ [2/3]**
> > >
> > > **2. Experiments on more challenging dataset and comprehensive evaluation on the transferability to different datasets and other downstream tasks. (further response to question 2,4,5)**
> > > > 2: The validation of the proposed method on more challenging datasets with higher resolution images is a must, since practical value is an important aspect for self-supervised learning methods.
> > >
> > > > 4: Authors should include a study on the transferability of distilled generator features, e.g. test the distilled features on a different test set from the training data.
> > >
> > > > 5: Other downstream tasks are needed to study the pros&cons of generator features as a self-supervised representation.
> > >
> > > As requested, we train and evaluate our method in more challenging datasets including ImageNet100 and ImageNet, and also report transfer performance of resulting models to various computer vision datasets. To this end, we first train a GAN and then distill its representations to the ResNet50 feature extractor using our “squeeze and span” on each dataset. Specifically, on ImageNet100, we train a Projected GAN [4] with StyleGAN2 architecture at 128x128 resolution until 25M images have been shown to the discriminator and obtain a pre-trained generator with 23.78 FID. On ImageNet, we use the pre-trained StyleGAN2 at 128x128 resolution that achieves FID 49.8, which is released by [5] (https://github.com/yandex-research/gan-transfer). The input resolution to the student network is set to 96x96, the batch size is set as 256, and the student model is trained for 100 epochs. For the baseline, we use VICReg and train it on the mixture of real and synthetic data by using the same configuration, i.e. input resolution 96x96, batch size 256, and 100 epochs.
> > >
> > > We evaluate the trained feature extractors in three aspects:
> > > (1) Linear classification performance on the test set of same dataset;
> > > (2) Linear classification performance on other datasets including Aircraft, Caltech101, Cars, CIFAR10, CIFAR100, DTD, Flowers, Food, Pets, SUN397, and VOC2007;
> > > (3) Downstream task finetuning on other tasks and datasets, including object detection on PASCAL VOC, surface normal estimation on NYUv2, and semantic segmentation on ADEChallenge2016.
> > > For the (2) & (3), we follow the protocol in [6] and use its released source code (https://github.com/linusericsson/ssl-transfer) to conduct evaluation. Please refer to [6] for details.
> > >
> > > The results are summarized in the following tables, where we could observe:
> > > - (1) As depicted in Table 2-1, our method outperforms the VICReg on the mixed data with a large margin on ImageNet100 (6.90% Top-1 Acc improvement) and a clear increase on ImageNet (0.48% Top-1 Acc improvement).
> > > - (2) As depicted in Table 2-2, our method achieves better transferability than VICReg on the mixed data no matter pre-trained on ImageNet100 or ImageNet. Our method beats VICReg on nearly all other datasets and the improvement on average accuracy is 3.40 with models pre-trained on ImageNet100 and 1.00 with models pre-trained on ImageNet.
> > > - (3) As depicted in Table 2-3, representations learned with our method can be well transferred to various downstream tasks such as object detection, surface normal estimation and semantic segmentation, and consistently show higher performance than VICReg.
> > >
> > > We believe these results suggest that generator features have strong transferability and great promise to contribute to self-supervised representation learning.

---

> > > ### Author Response · Authors · 2022-08-09
> > > **Further Response to Reviewer pYVQ [1/3]**
> > >
> > > **1. Experiments with different GAN architectures (further response to question 3)**
> > > > 3: Studying the generator features of different GAN architectures is important for studying the effectiveness of generator features since different architectures naturally have different inductive bias.
> > >
> > > As requested, we train and evaluate our method with the generator features that are obtained from different GAN architectures including AutoGAN [1], StyleGAN-XL [2], and BigGAN [3] on CIFAR10, and report the results in the table below. In particular, we report results with our “squeeze” model instead of our full method “squeeze and span” to investigate the impact of synthetic data and generator feautre quality on distilling representations without the real data. We use the same feature selection strategy across different architectures as before: the consecutive network layers are grouped into a block according to its output resolution and the output feature maps of block at each resolution are chosen (Section B.2 in the appendix).
> > >
> > > In the table below, we report whether the GAN is pretrained in conditional or unconditional manner, the link to publicly available pre-trained checkpoints, and their FID. The best results are in **bold** and the second best is *italic*. From these results, we would like to highlight the following points:
> > >
> > > - (1) Our method yields good performance with prevalent GAN architectures such as StyleGAN2, StyleGAN3, and BigGAN, showing that our method is architecture-agnostic.
> > >
> > > - (2) When the generators are conditioned on class labels, they result in better distillation and hence classification performance than unconditional ones, possibly due to its higher generation quality (conditional StyleGAN2-ADA achieves 0.5 FID lower than unconditional one) and the embedded class information in conditional modeling. Since training conditional GAN requires class labels which violates the goal of unsupervised learning, we only report these results for sake of completeness.
> > >
> > > - (3) Although StyleGAN-XL with StyleGAN3 architecture achieves the highest generation quality (1.85 FID), its distillation performance is 2.70 top-1 acc lower than StyleGAN2. This suggests that generation quality may not be the only factor determining the representation transfer ability. We hope to further study this problem in the future.
> > >
> > > | GAN Arch | Type | Sources | FID | Squeeze Top-1 Acc |
> > > | :---: | :---: | :---: | :---: | :---: |
> > > | StyleGAN2-ADA | Unconditional | https://github.com/NVlabs/stylegan2-ada-pytorch | 2.92 | *87.67* |
> > > | AutoGAN | Unconditional | https://github.com/VITA-Group/AutoGAN | 12.42 | 76.28 |
> > > | StyleGAN2-ADA | Conditional | https://github.com/NVlabs/stylegan2-ada-pytorch |  *2.42* | **88.90** |
> > > | StyleGAN-XL (StyleGAN3) | Conditional | https://github.com/autonomousvision/stylegan_xl | **1.85** | 84.97 |
> > > | BigGAN-DiffAugment-cr | Conditional | https://github.com/mit-han-lab/data-efficient-gans/tree/master/DiffAugment-biggan-cifar | 8.49 | 86.41 |
> > >
> > > Table1-1. Squeeze performance with different GAN architectures on CIFAR10.

---

> ### Author Response · Authors · 2022-08-02
> **Response to Reviewer pYVQ [1/2]**
>
> **1. One missing related work. Generative Hierarchical Features from Synthesizing Images, CVPR 2021.**
>
> GHFeat [1] maps a given image back into the style codes of StyleGAN to reconstruct the given image and take the reconstructed style codes as image representations. We think that this work can be categorized as “encoding” in our Fig.1(b). Unlike them that learns representation through image reconstruction, our work is built upon the representation distillation, c.f. Fig.1(c).
> Thanks for the constructive reminder. The citation and discussion about this work are included in the updated manuscript. (Line25, and Line 66~67).
> > … GHFeat [58] trains a post hoc encoder that maps given images back into style codes of style-based GANs [35, 36, 34] and takes the style codes as image representations. …
>
> **2. The validation of the proposed method on more challenging datasets with higher resolution images is a must, since practical value is an important aspect for self-supervised learning methods.**
>
> We have justified the effectiveness of our method on CIFAR10, CIFAR100, STL10 benchmarks which are also widely considered in SSL [2, 3]. We are currently running more experiments on ImageNet-like datasets and hope to post them in the discussion period. Please see the response to @Reviewer 8CJZ for ImageNet100 results.
>
> **3. Studying the generator features of different GAN architectures is important for studying the effectiveness of generator features since different architectures naturally have different inductive bias.**
>
> Thanks for the suggestion. It is worth noting that the proposed squeeze and span method is materialized in a GAN architecture-agnostic fashion pursuing generator feature distillation, Fig 3 (in the manuscript). Our approach is built on StyleGAN2(-ADA) for its top performance in generation quality, training stability, and versatility (Line 225~226). We believe the superior performances presented in Table 2 have justified the efficacy of our method. Still, We are running experiments for various architectures including AutoGAN [4], StyleGAN3 [5], and BigGAN [6], and hope to include more results in the discussion period.
>
> **4. Authors should include a study on the transferability of distilled generator features, e.g. test the distilled features on a different test set from the training data.**
>
> Thanks for the suggestion. We have conducted cross-dataset linear classification evaluation over CIFAR10, CIFAR100, and STL10 for our trained models. Results are listed as below, where the first and second rows show source and target datasets respectively. One can see that our method (Sq & Sp) typically achieves better cross-dataset generalization compared to SimSiam and VICReg.
>
> | Pretrain Data | Methods | CIFAR10 | | CIFAR100 | | STL10 | |
> | :---: | :---: | :---: | :---: | :---: | :---: | :---: | :---: |
> | | | CIFAR100 | STL10 | CIFAR10 | STL10 | CIFAR10 | CIFAR100 |
> | Real & Syn | SimSiam | 41.32 | 68.70 | 75.90 | 60.41 | 59.34 | 32.88 |
> | Real & Syn | VICReg | 54.19 | 80.56 | 79.11 | **74.73** | 58.15 | 31.25 |
> | Real & Syn | Sq & Sp (Ours) | **58.93** | **82.22** | **80.68** | 73.85 | **64.76** | **37.48** |

---

### Official Review · Reviewer_qg1m · 2022-07-12

**Rating:** 5
**Confidence:** 3
**Soundness:** 2 fair
**Presentation:** 2 fair
**Contribution:** 2 fair

**Summary:**

The main idea of the paper is how to use the pre-trained generator network of GAN to improve the downstream task of classification. Specifically, the paper use with VICReg-based method to train the network S (called student) that takes the real inputs and applies different augmentations before putting them as inputs of S, the features output of these augmentations is regularized to be similar (invariant to transformation) while the variance-covariance are maximum (this task is called “span” module). Then, the method applies the pre-trained generator to synthesize images and these images are input into the S to produce the features which are regularized from match with those features from concatenation of feature maps (go through module called “squeeze” module before that) that generated the synthetic image. The losses for synthetic images are similar to on real images except the distilled loss between generator’s features and output of the student' feature. The experiments on low resolution datasets (CIFAR-10, CIFAR-100 and STL10) showing the improvements over two existing works SiamSiam and VICReg trained on the real, fake or mixed data. The paper conducted the ablation study to understand the contributions of different components.

However, the paper is written by the other way around: starting with “squeeze” and improve with “span” module that I think want to make the idea more interesting (but probably makes reader a bit confusing on what is actual contribution)?  With this order, the paper investigates different ways to exact features from GAN: from discriminator, latent variable and generator. The 2D visualisation and experiments suggest the features from generator are most discriminative. Inspired from the generator’s features, the paper introduces two modules “squeeze” and “span” to extract the generator features for distillation.

**Questions:**

See above

**Limitations:**

Yes

**Strengths And Weaknesses:**


**Strength**

I think the main contribution this paper regarding the "squeeze" module and how the generator being used to improve the existing self-supervised models. It is novel enough to me and this brings some improvements (2%) from the simple way of mixing real/fake dataset. The paper is well-written and easy to read.

**Weakness**

From my perspective, “span” module is considered as one contribution is questionable as it is just similar to VICReg method. Therefore, I am not sure some claims of the span module to address the mode collapse and the gap between real and fake is convincing enough, except if the authors can bring some theoretical / empirical results to show that. Otherwise, it is not highly motivated why we need the generator for distillation instead of the classifier?

At the moment, I put as borderline accept but the rating can be increased and decreased depends also comments of other reviewers which may point out something I am missing and also the rebuttal.

---

> ### Author Response · Authors · 2022-08-02
> **Response to Reviewer qg1m**
>
> **1. "span" module as one contribution is questionable as it is similar to VICReg.**
>
> Our “span” module is motivated to reduce real versus synthetic domain gap on student performance. Hence, the key contribution of “span” lies more in the identification and simple solution to this problem. Though we instantiate the span module with VICReg, we think other techniques such as distribution alignment [1] may also help to mitigate this issue, but more implementation plans are beyond the scope of this paper. Overall, we claim “span” as one contribution mainly from the perspective of motivation, simple solution and its effect, not from the technique perspective.
>
> **2. It is unclear if the span module address the mode collapse and domain gap between real and fake images without theoretical / empirical results provided. Otherwise, It is not highly motivated why we need the generator for distillation instead of the classifier?**
>
> Thanks for the detailed comments. The main of concern here is the effect of “span” module on domain gap issue, which also raises the question on its contribution. We would like to summarize the following empirical results:
>  - A. The existence of domain gap is evidenced in Table 2 by that “Both SimSiam and VICReg perform worse when pre-trained on only synthetic data than only real data …” (Line 264 ~ Line 266).
> - B.  we provide additional empirical results below to show that domain gap is reduced with our method. To make our point, we train four different networks: (1) VICReg on synthetic images, (2) VICReg on a mixture of real and synthetic images, (3) “Squeeze” on synthetic images, (4) “Squeeze and Span” on a mixture of real and synthetic images. We use the learned representation network to extract features of synthetic images and real images and compute the squared MMD between these two bank of representation. The squared MMD is computed with polynomial kernel similarly to the KID metric [2] which can suggest the gap between two domains. Results listed below show “squeeze and span” obtains significantly lower squared MMD than “squeeze”, suggesting that our span module can help to reduce the domain gap.
>
> | Method | Pre-train Domain | CIFAR10 | CIFAR100 | STL10 |
> | :---: | :---: | :---: | :---: | :---: |
> | VICReg | Syn | 3.44 e-5 | 5.89 e-5 | 5.39 e-3 |
> | VICReg | Real & Syn | 3.74 e-5 | 16.8 e-5 | 11.4 e-3 |
> | Squeeze | Syn | 4.79 e-5 | 1.24 e-5 | 9.82 e-3 |
> | Squeeze and Span | Real & Syn | 0.45 e-5 | 0.25 e-5 | 3.71 e-3 |
>
> - C. In Table 2 in the manuscript, “Squeeze and Span” significantly improve the student performance compared to “Span” module, showing that after narrowing the domain gap, our method can improve the student performance.
>
> These empricial results justify the existence of domain gap issue and our method successfully mitigate this issue and accordingly improve the overall performance.
>
> **Reference**
> - [1] Yaroslav Ganin, Victor Lempitsky. Unsupervised Domain Adaptation by Backpropagation. In ICML, 2015.
> - [2] Mikołaj Bińkowski, Danica J. Sutherland, Michael Arbel, Arthur Gretton. Demystifying mmd gans. In ICLR, 2018.

---

### Author Response · Authors · 2022-08-02
**Revision Summary (by 2 Aug)**

We thank all the reviewer for their valuable feedback!

Reviewers have identified the strength of our paper as significant academic value [$\textcolor{blue}{@Reviewer}$ $\textcolor{blue}{pYVQ}$], well-motivated idea [$\textcolor{blue}{@Reviewer}$ $\textcolor{blue}{pYVQ}$, $\textcolor{green}{@Reviewer}$ $\textcolor{green}{8CJZ}$], simple method [$\textcolor{orange}{@Reviewer}$ $\textcolor{orange}{qg1m}$], effective results [$\textcolor{orange}{@Reviewer}$ $\textcolor{orange}{qg1m}$, $\textcolor{green}{@Reviewer}$ $\textcolor{green}{8CJZ}$], and good writing [$\textcolor{orange}{@Reviewer}$ $\textcolor{orange}{qg1m}$,  $\textcolor{blue}{@Reviewer}$ $\textcolor{blue}{pYVQ}$, $\textcolor{green}{@Reviewer}$ $\textcolor{green}{8CJZ}$].

Reviewers also raise main concerns about the clarification of “span” module and additional experiments.

Our paper and appendix have been revised accordingly. The revised part is marked as $\textcolor{red}{red}$. In particular, we have made the following change:

I. Clarificationof “span” module:
 - Empricial results that domain gap issue is mitigated (Line 286 289, Table 4 in the main paper) [$\textcolor{orange}{@Reviewer\;qg1m}$]

II. Additional experiements
- Results of “squeeze” and “span” for latent variables (Table 1 in the main paper) [$\textcolor{green}{@Reviewer}$ $\textcolor{green}{8CJZ}$]
- ImageNet100 evaluation (Table 2 in the main paper; Line 268 273) [ $\textcolor{blue}{@Reviewer}$ $\textcolor{blue}{pYVQ}$, $\textcolor{green}{@Reviewer}$ $\textcolor{green}{8CJZ}$]
- Cross-dataset evaluation (Line16 22 in the appendix) [ $\textcolor{blue}{@Reviewer}$ $\textcolor{blue}{pYVQ}$]
- Ablation analysis about the choice of generation blocks (Line 23 42 in the appendix) [ $\textcolor{blue}{@Reviewer}$ $\textcolor{blue}{pYVQ}$]

III. Minor revision
- Discussion about related work GHFeat (Line66 67 in the main paper) [ $\textcolor{blue}{@Reviewer}$ $\textcolor{blue}{pYVQ}$]
- Updated STL10 evaluation (Table2 in main paper). Our previous configuration of linear classification evaluation on STL10 leads to insufficient training of the linear classifier and low reported performance. The configuration has been corrected to “90 epochs using SGD with a base learning rate of 30.0, momentum 0.9, weight decay 0., and batch size 256” as described in Line 237 239 in the main paper.

---

> ### Author Response · Authors · 2022-08-09
> **Further Revision (by 9 Aug)**
>
> During the discussion period, we have further revised our submission to include additional experiment results which we hope can further address the concerns of $\textcolor{blue}{@Reviewer}$ $\textcolor{blue}{pYVQ}$ about experiments.
>
> - Experiments with different GAN architectures. (Section B.3 in the appendix) [$\textcolor{blue}{@Reviewer}$ $\textcolor{blue}{pYVQ}$]
> - Experiments on ImageNet. (Table2 and line 272--276 in the main text) [$\textcolor{blue}{@Reviewer}$ $\textcolor{blue}{pYVQ}$]
> - Comprehensive evaluation of transferability to other 11 classification datasets and 3 downstream tasks. (Section B.1 in the appendix) [$\textcolor{blue}{@Reviewer}$ $\textcolor{blue}{pYVQ}$]
>
> Given the updated experiment evaluation results, our conclusion and limitation are also updated accordingly.

---

### Meta-Review · Area_Chair_ApPM · 2022-08-28

**Recommendation:** Accept
**Confidence:** Certain

**Metareview:**

While there were initial concerns about the tasks used for evaluation, the authors did extensive extra experiments and satisfied many of these concerns.

**Award:**

No

---

### Decision · Program_Chairs · 2022-09-14

Accept